# Distributed Adversarial Training to Robustify Deep Neural Networks at Scale

**Gaoyuan Zhang**[1,*]    **Songtao Lu**[1,*]    **Yihua Zhang**[2]    **Xiangyi Chen**[3]    **Pin-Yu Chen**[1]    **Quanfu Fan**[1]    **Lee Martie**[1]

**Lior Horesh**[1]                **Mingyi Hong**[3]                **Sijia Liu**[1,2]

[1]IBM Research
[2]Michigan State University
[3]University of Minnesota
[*]Equal Contribution

## Abstract

Current deep neural networks (DNNs) are vulnerable to adversarial attacks, where adversarial perturbations to the inputs can change or manipulate classification. To defend against such attacks, an effective and popular approach, known as *adversarial training (AT)*, has been shown to mitigate the negative impact of adversarial attacks by virtue of a min-max robust training method. While effective, it remains unclear whether it can successfully be adapted to the distributed learning context. The power of distributed optimization over multiple machines enables us to scale up robust training over large models and datasets. Spurred by that, we propose *distributed adversarial training (DAT)*, a *large-batch* adversarial training framework implemented over multiple machines. We show that DAT is general, which supports training over labeled and unlabeled data, multiple types of attack generation methods, and gradient compression operations favored for distributed optimization. Theoretically, we provide, under standard conditions in the optimization theory, the convergence rate of DAT to the first-order stationary points in general non-convex settings. Empirically, we demonstrate that DAT either matches or outperforms state-of-the-art robust accuracies and achieves a graceful training speedup (e.g., on ResNet–50 under ImageNet). Codes are available at https://github.com/dat-2022/dat.

## 1 INTRODUCTION

The rapid increase of research in DNNs and their adoption in practice is, in part, owed to the significant breakthroughs made with DNNs in computer vision [Alom et al., 2018]. Yet, with the apparent power of DNNs, there remains a se- rious weakness of robustness. That is, DNNs can easily be manipulated (by an adversary) to output drastically different classifications and can be done so in a controlled and directed way. This process is known as an adversarial attack and considered as one of the major hurdles in using DNNs in security critical and real-world applications [Goodfellow et al., 2015, Szegedy et al., 2013, Carlini and Wagner, 2017, Papernot et al., 2016, Kurakin et al., 2016, Xu et al., 2019b].

Methods to train DNNs being robust against adversarial attacks are now a major focus in research [Xu et al., 2019a]. But most of them are far from satisfactory [Athalye et al., 2018] with the exception of the adversarial training (AT) approach [Madry et al., 2017]. AT is a min-max robust training method that minimizes the worst-case training loss at adversarially perturbed examples. AT has inspired a wide range of state-of-the-art defenses [Zhang et al., 2019b, Sinha et al., 2018, Boopathy et al., 2020, Carmon et al., 2019, Shafahi et al., 2019, Zhang et al., 2019a], which ultimately resort to min-max optimization. However, different from standard training, AT is more computationally intensive and is difficult to scale.

**Motivation and challenges.**    *First*, although a 'fast' version of AT (we call Fast AT) was developed in [Wong et al., 2020] where an iterative inner maximization solver is replaced by a simplified (single-step) solution, it may suffer several problems compared to AT: unstable robust learning performance [Li et al., 2020], over-sensitive to learning rate schedule [Rice et al., 2020], and catastrophic forgetting of robustness against strong attacks [Andriushchenko and Flammarion, 2020]. As a result, AT is still the dominant robust training protocol across applications. Spurred by that, we propose DAT, a new approach to speed up AT by allowing for scaling batch size with distributed machines. *Second*, existing AT-type methods are generally built on *centralized* optimization. The need of AT in a *distributed* setting arises when centralized robust training becomes infeasible or ineffective. For example, training data are distributed as they cannot centrally be stored at a single machine due to their

*Accepted for the 38th Conference on Uncertainty in Artificial Intelligence* (UAI 2022).

size or privacy. Or computing units are distributed as they allow large-batch optimization to improve the scalability of training.

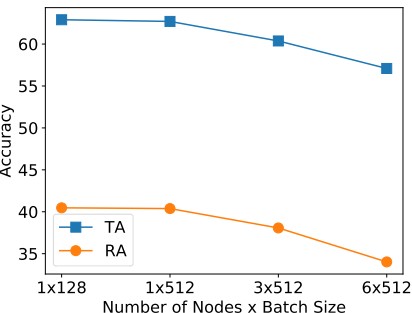

Figure 1: Robust accuracy (RA) and standard test accuracy (TA) of AT vs. scaled batch size under (ImageNet, ResNet-50) using distributed machines.

While designing a distributed solution is important, doing so effectively is non-trivial. Figure 1 demonstrates an example: When scaling batch size with the number of computing nodes, the conventional AT method yields a large performance drop in both robust and standard accuracies. Thus, the adaptation of AT to distributed learning leaves many unanswered questions. In this work, we aim to design a principled and theoretically-grounded (large-batch) DAT framework by making full use of the computing capability of multiple data-locality (distributed) machines, and show that DAT expands the capacity of data storage and the computational scalability. Furthermore, due to the existence of many variants of AT, it requires a careful and systematic study on distributed AT in its formulation, methodology, theory and performance evaluation.

**Contributions.**    We list our main contributions below.

*(i)* We provide a general algorithmic framework for DAT, which supports multiple (large-batch) distributed variants of AT, e.g., supervised AT and semi-supervised AT.

*(ii)* In theory, we quantify how descent errors from multiple sources (gradient estimation, quantization, adaptive learning rate, and inner maximization oracle) affect the convergence of DAT. We prove that the convergence speed of DAT to the first-order stationary points in general non-convex settings at a rate of $O(1/\sqrt{T})$, where $T$ is the total number of iterations. This result matches the standard convergence rate of classic training algorithms, e.g., stochastic gradient descent (SGD), for only the minimization problems.

*(iii)* In practice, we make a comprehensive empirical study on DAT, showing its effectiveness to (1) robust training over ImageNet, (2) provably robust training by randomized smoothing, (3) robust training with unlabeled data, (4) robust pretraining + finetuning, and (5) robust training across different computing and communication configurations.

## 2  RELATED WORK

**Training robust classifiers.**    AT [Madry et al., 2017], the first known min-max optimization-based defense, has inspired a wide range of other effective defenses. Examples include adversarial logit pairing [Kannan et al., 2018], input gradient or curvature regularization [Ross and Doshi-Velez, 2018, Moosavi-Dezfooli et al., 2019], trade-off between robustness and accuracy (TRADES) [Zhang et al., 2019b], distributionally robust training [Sinha et al., 2018], dynamic adversarial training [Wang et al., 2019b], robust input attribution regularization [Boopathy et al., 2020], certifiably robust training [Wong and Kolter, 2017], and semi-supervised robust training [Stanforth et al., 2019, Carmon et al., 2019].

In particular, some recent works proposed *fast but approximate* AT algorithms, such as 'free' AT [Shafahi et al., 2019], you only propagate once (YOPO) [Zhang et al., 2019a], and fast gradient sign method (FGSM) based AT [Wong et al., 2020]. These algorithms achieve speedup in training by simplifying the inner maximization step of AT, but are designed for centralized model training. A few works made empirical efforts to scale AT up by using multiple computing nodes [Xie et al., 2019, Kang et al., 2019, Qin et al., 2019], they were limited to specific use cases and lacked a thorough study on when and how distributed learning helps, either in theory or in practice.

**Distributed model training.**    Distributed optimization has been found to be effective for the standard training of machine learning models [Dean et al., 2012, Goyal et al., 2017, You et al., 2019, Chen et al., 2020]. In contrast to centralized optimization, distributed learning enables increasing the batch size proportional to the number of computing nodes/machines. However, it is challenging to train a model via large-batch optimization without incurring accuracy loss compared to the standard training with same number of epochs [Krizhevsky, 2014, Keskar et al., 2017]. To tackle this challenge, it was shown in [You et al., 2017b, 2018, 2019] that adaptation of learning rates to the increase of the batch size is an essential mean to boost the performance of large-batch optimization. A layer-wise adaptive learning rate strategy was then proposed to speed up the training as well as preserve the accuracy. Although these works have witnessed several successful applications of distributed learning in training *standard* image classifiers, they leave the question of how to build *robust* DNNs with DAT open. In this paper, we show that the power of layer-wise adaptive learning rate also applies to DAT. Since distributed learning introduces machine-machine communication overhead, another line of work [Alistarh et al., 2017, Yu et al., 2019, Bernstein et al., 2018, Wangni et al., 2018, Stich et al., 2018, Wang et al., 2019a] focused on the design of communication-efficient distributed optimization algorithms.

The study on distributed learning is extensive, but the prob-

lem of distributed min-max optimization is less explored, with some exceptions [Srivastava et al., 2011, Notarnicola et al., 2018, Tsaknakis et al., 2020, Liu et al., 2019a,b]. A key difference to our work is that none of the aforementioned literature studied the *large-batch min-max optimization* with its applications to training *robust* DNNs, neither theoretically nor empirically. While there are recent proposed algorithms for training Generative Adversarial Nets (GANs) [Liu et al., 2019a,b], training robust DNNs against adversarial examples is intrinsically different from GAN training. In particular, training robust DNNs requires inner maximization with respect to each training data rather than empirical maximization with respect to model parameters. Such an essential difference leads to different optimization goals, algorithms, convergence analyses and implementations.

## 3 PROBLEM FORMULATION

In this section, we first review the standard setup of adversarial training (AT) [Madry et al., 2017], and then propose a general min-max setup for distributed AT (DAT).

**Adversarial training.** AT [Madry et al., 2017] is a min-max optimization method for training robust ML/DL models against adversarial examples [Goodfellow et al., 2015]. Formally, AT solves the problem

$$\underset{\boldsymbol{\theta}}{\text{minimize}} \; \mathbb{E}_{(\mathbf{x},y)\in\mathcal{D}} \left[ \underset{\|\boldsymbol{\delta}\|_\infty \leq \epsilon}{\text{maximize}} \; \ell(\boldsymbol{\theta}, \mathbf{x}+\boldsymbol{\delta}; y) \right], \quad (1)$$

where $\boldsymbol{\theta} \in \mathbb{R}^d$ denotes the vector of model parameters, $\boldsymbol{\delta} \in \mathbb{R}^n$ is the vector of input perturbations within an $\ell_\infty$ ball of the given radius $\epsilon$, namely, $\|\boldsymbol{\delta}\|_\infty \leq \epsilon$, $(\mathbf{x}, y) \in \mathcal{D}$ corresponds to the training example $\mathbf{x}$ with label $y$ in the dataset $\mathcal{D}$, and $\ell$ represents a pre-defined training loss, e.g., the cross-entropy (CE) loss. The rationale behind problem (1) is that the model $\boldsymbol{\theta}$ is robustly trained against the *worst-case* loss induced by the adversarially perturbed samples. It is worth noting that the AT problem (1) is *different* from conventional stochastic min-max optimization problems, e.g., GANs training [Goodfellow et al., 2014]. Note that in (1), the stochastic sampling corresponding to the expectation over $(\mathbf{x}, \mathbf{y}) \in \mathcal{D}$ is conducted *prior to* the inner maximization operation. Such a difference leads to the *sample-specific* adversarial perturbation $\boldsymbol{\delta}(\mathbf{x}) := \text{maximize}_{\|\boldsymbol{\delta}\|_\infty \leq \epsilon} \; \ell(\boldsymbol{\theta}, \mathbf{x}+\boldsymbol{\delta}; y)$.

**Distributed AT (DAT).** Let us consider a popular parameter-server model of distributed learning [Dean et al., 2012]. Formally, there exist $M$ workers each of which has access to a local dataset $\mathcal{D}^{(i)}$, and thus $\mathcal{D} = \cup_{i=1}^M \mathcal{D}^{(i)}$. There also exists a server/master node (e.g., one of workers could perform as server), which collects local information

(e.g., individual gradients) from the other workers to update the model parameters $\boldsymbol{\theta}$. Spurred by (1), DAT solves problems of the following generic form,

$$\underset{\boldsymbol{\theta}}{\text{minimize}} \; \frac{1}{M} \sum_{i=1}^M f_i(\boldsymbol{\theta}; \mathcal{D}^{(i)}),$$
$$f_i =: \mathbb{E}_{(\mathbf{x},y)\in\mathcal{D}^{(i)}} \left[ \lambda\ell(\boldsymbol{\theta}; \mathbf{x}, y) + \max_{\|\boldsymbol{\delta}\|_\infty \leq \epsilon} \phi(\boldsymbol{\theta}, \boldsymbol{\delta}; \mathbf{x}, y) \right] \quad (2)$$

where $f_i$ denotes the local cost function at the $i$th worker, $\phi$ is a robustness regularizer against the input perturbation $\boldsymbol{\delta}$, and $\lambda \geq 0$ is a regularization parameter that strikes a balance between the training loss and the worst-case robustness regularization. In (2), if $M = 1$, $\mathcal{D}^{(1)} = \mathcal{D}$, $\lambda = 0$ and $\phi = \ell$, then the DAT problem reduces to the AT problem (1). We cover two categories of (2). ① DAT with labeled data: In (2), we consider $\phi(\boldsymbol{\theta}, \boldsymbol{\delta}; \mathbf{x}, y) = \ell(\boldsymbol{\theta}, \mathbf{x} + \boldsymbol{\delta}; y)$ with labeled training data $(\mathbf{x}, y) \in \mathcal{D}^{(i)}$ for $i \in [M]$. Here $[M]$ denotes the integer set $\{1, 2, \ldots, M\}$. ② DAT with unlabeled data: In (2), different from DAT with labeled data, we augment $\mathcal{D}^{(i)}$ with an unlabeled dataset, and define the robust regularizer $\phi$ as the pseudo-labeled worst-case CE loss [Carmon et al., 2019] or the TRADES regularizer [Stanforth et al., 2019, Zhang et al., 2019b].

## 4 METHODOLOGIES

At the first glance, distributed learning seems being naturally applied since problem (2) is decomposable over multiple workers. Yet, the actual case is much more complex. **First**, in contrast to standard AT, DAT allows for using a $M$ times larger batch size to update the model parameters $\boldsymbol{\theta}$ in (2). Thus, given the same number of epochs, DAT takes $M$ fewer gradient updates than AT. Although there exist some large-batch model training techniques for solving *min-only* problems [You et al., 2017a,b, 2018, 2019, Goyal et al., 2017, Keskar et al., 2017], it remains unclear if they are effective to DAT due to its *min-max* optimization nature. **Second**, either AT or distributed learning has its own challenges. In AT, for ease of attack generation, i.e., conducting inner maximization of (2), fast gradient sign method (FGSM) was leveraged to improve its computation efficiency [Wong et al., 2020]. In distributed learning, gradient compression [Alistarh et al., 2017, Yu et al., 2019] was used for reducing communication overhead. Thus, it also remains unclear whether these customizations are adaptable to DAT. In a nutshell, the distributed min-max optimization-based robust training algorithm has not been well studied previously, particularly in the use of different types of attack generators (inner maximization oracles), gradient quantization, large-batch size, and adaptive learning rate. Although either of the standalone techniques was studied separately, justifying their coherent integration 'actually works' (both practically and theoretically) is quite demanding.

**Algorithmic framework of DAT.** DAT follows the framework of distributed learning with parameter server. In what follows, we elaborate on its key components through its meta-form shown by Algorithm 1 (see its detailed version in Algorithm A1). DAT contains three algorithmic blocks. In the *first* block, every distributed worker calls for a maximization oracle to obtain the adversarial perturbation for each sample within a data batch, then computes the gradient of the local cost function $f_i$ in (2) with respect to (w.r.t.) model parameters $\boldsymbol{\theta}$. And every worker is allowed to quantize/compress the local gradient prior to transmission to the server. In the *second* block, the server aggregates the local gradients, and transmits the aggregated gradient (or the quantized gradient) to the other workers. In the *third* block, the model parameters are eventually updated by a minimization oracle at each worker based on the received gradient information from the server.

---

**Algorithm 1** Meta-version of DAT (Alg. A1 in Supplement)

---

1: **for** Worker $i = 1, 2, \ldots, M$ **do**         ▷ Block 1
2:     Sample-wise attack generation (A1)
3:     Local gradient computation (A2)
4:     Worker-server communication
5: **end for**
6: Gradient aggregation at server (A3)         ▷ Block 2
7: Server-worker communication
8: **for** Worker $i = 1, 2, \ldots, M$ **do**         ▷ Block 3
9:     Model parameter update (A4)
10: **end for**

---

**Large-batch challenge in DAT and a layerwise adaptive learning rate (LALR) solution.** In DAT, the aggregated gradient (Step 6 in Algorithm 1) is built on the data batch that is $M$ times larger than the standard AT. This leads to a large-batch challenge in min-max optimization. This challenge can also be verified from Fig. 1. To overcome the large-batch challenge, we adopt the technique of layerwise adaptive learning rate (LALR), backed up by the recent successful applications to the standard training of large-scale image classification and language modeling networks with large data batch [You et al., 2019, 2017b].

To be more specific, the model training recipe using LALR becomes

$$\boldsymbol{\theta}_{t+1,i} = \boldsymbol{\theta}_{t,i} - \frac{\tau(\|\boldsymbol{\theta}_{t,i}\|_2) \cdot \eta_t}{\|\mathbf{u}_{t,i}\|_2} \cdot \mathbf{u}_{t,i}, \quad \forall i \in [h], \quad (3)$$

where $\boldsymbol{\theta}_{t,i}$ denotes the $i$th-layer parameters at iteration $t$, with $\boldsymbol{\theta}_t = [\boldsymbol{\theta}_{t,1}^\top, \ldots, \boldsymbol{\theta}_{t,h}^\top]^\top$, $h$ is the number of layers, $\mathbf{u}_t$ is a descent direction computed based on the first-order gradient w.r.t. model parameters $\boldsymbol{\theta}_t$, $\tau(\|\boldsymbol{\theta}_{t,i}\|_2) = \min\{\max\{\|\boldsymbol{\theta}_{t,i}\|_2, c_l\}, c_u\}$ is a *layerwise* scaling factor of the *adaptive* learning rate $\frac{\eta_t}{\|\mathbf{u}_{t,i}\|_2}$, and $c_l = 0$ and $c_u = 10$

are set in our experiments (see Appendix 4.1 for some ablation studies on hyperparameter selection).

In (3), the specific form of the descent direction $\mathbf{u}_t$ is determined by the optimizer employed. For example, if the adaptive momentum (Adam) method is used, then $\mathbf{u}_t$ is given by the exponential moving average of past gradients scaled by square root of exponential moving averages of squared past gradients [Reddi et al., 2018, Chen et al., 2018]. Such a variant of (3) that uses Adam as the base algorithm is also known as LAMB [You et al., 2019] in standard training. However, it was elusive if the advantage of LALR is preserved in large-batch min-max optimization. As will be evident later, the effectiveness of LALR in DAT can be justified from both theoretical and empirical perspectives. The rationale is that the layer-wise adaptive learning rate smooths the optimization trajectory so that a larger learning rate can be used without causing sharp optima even in distributed min-max optimization.

**Other add-ons for DAT.** In what follows, we illustrate two add-ons to improve computation and communication efficiency of DAT.

➢ *Inner maximization: Iterative vs. one-shot solution.* In DAT, each worker calls for an inner maximization oracle to generate adversarial perturbations (Step 2 of Algorithm 1). We specify two solvers of perturbation generation: iterative projected gradient descent (PGD) and one-shot (projected) FGSM [Goodfellow et al., 2015, Wong et al., 2020]. Our experiments will show that FGSM together with LALR works well in DAT. We also remark that other techniques [Shafahi et al., 2019, Zhang et al., 2019a] can also be used to simplify inner maximization, however, we focus on FGSM since it is computationally lightest.

➢ *Gradient quantization.* In contrast to standard AT, DAT may call for worker-server communications (Steps 4 and 7 of Algorithm 1). That is, if a single-precision floating-point data type is used, then DAT needs to transmit $32d$ bits per worker-server communication at each iteration. Recall that $d$ is the dimension of $\boldsymbol{\theta}$. In order to reduce the communication cost, DAT has the option to quantize the transmitted gradients using a fixed number of bits fewer than 32. We specify the gradient quantization operation as the *randomized quantizer* [Alistarh et al., 2017, Yu et al., 2019]. In Sec. 6 we will show that DAT, combined with gradient quantization, still leads to a competitive performance. For example, the robust accuracy of ResNet-50 trained by a 8-bit DAT (performing quantization at Step 4 of Algorithm 1) for ImageNet is just $0.55\%$ lower than the robust accuracy achieved by the 32-bit DAT. It is also worth mentioning that the All-reduce communication protocol can be regarded as a special case of the parameter-server setting in Algorithm 1 when every worker performs as a server. In this case, the communication network becomes fully connected and only the worker-server communication (Step 4 of Algorithm 1)

is needed. Please refer to Appendix 1 for more details on gradient quantization.

# 5 CONVERGENCE ANALYSIS OF DAT

Although standard AT has been proved with convergence guarantees [Wang et al., 2019b, Gao et al., 2019], none of existing work addressed the convergence of DAT and took into account LALR and gradient quantization, even in the standard AT setup. Different from AT, DAT needs to quantify the descent errors from multiple sources (such as gradient estimation, quantization, adaptive learning rate, and inner maximization oracle). Before showing the challenges of proving the convergence rate guarantees, we first give the following assumptions.

**Assumptions.** Defining $\Psi(\boldsymbol{\theta}) := \frac{1}{M} \sum_{i=1}^{M} f_i(\boldsymbol{\theta}; \mathcal{D}^{(i)})$ in (2), we measure the convergence of DAT by the first-order stationarity of $\Psi$. Prior to convergence analysis, we impose the following assumptions: ($\mathcal{A}1$) $\Psi(\boldsymbol{\theta})$ is with layer-wise Lipschitz continuous gradients; ($\mathcal{A}2$) $\phi$ in (2) is strongly concave with respect to $\boldsymbol{\delta}$ and with Lipschitz continuous gradients within the perturbation constraint; ($\mathcal{A}3$) Stochastic gradient is unbiased and has bounded variance for each worker denoted by $\sigma^2$. Note that the validity of ($\mathcal{A}2$) could be justified from [Sinha et al., 2018, Wang et al., 2019b] by imposing a strongly convex regularization into the neighborhood of $\boldsymbol{\delta}$. $\mathcal{A}2$ is needed for tractability of analysis. We refer readers to Appendix 2.1 for more justifications on our assumptions ($\mathcal{A}1$)-($\mathcal{A}3$).

**Technical challenges.** In theory, the incorporation of LALR makes the analysis of min-max optimization highly non-trivial. The fundamental challenge lies in the nonlinear coupling between the biased adaptive gradient estimate resulted from LALR and the additional error generated from alternating update in DAT. From (3), we can see that the updated $\boldsymbol{\theta}$ is based on the normalized gradient, while if we perform convergence by applying the gradient Lipschitz continuity, the descent of the objective is measured by $\nabla\Psi(\boldsymbol{\theta}_t)$. This mismatch in the magnitude results in the bias term. The situation here is even worse, since the maximization problem cannot be solved exactly, the size of the bias depends on how close between the output of the oracle and the optimal solution w.r.t. $\boldsymbol{\delta}$ given $\boldsymbol{\theta}$.

We have proposed a new descent lemma (Lemma 2 in Appendix) to measure the decrease of the objective value in the context of alternative optimization, and showed that the bias error resulted from the layer-wise normalization can be compensated by large-batch training (Theorem 1). Prior to our work, we are *not* aware of any established convergence analysis for large-batch min-max optimization.

**Convergence rate.** In Theorem 1, we present the sublinear rate of DAT.

**Theorem 1.** *Suppose that assumptions $\mathcal{A}1$-$\mathcal{A}3$ hold, the inner maximizer of DAT provides a $\varepsilon$-approximate solution (i.e., the $\ell_2$-norm of inner gradient is upper bounded by $\varepsilon$), and the learning rate is set by $\eta_t \sim \mathcal{O}(1/\sqrt{T})$, then $\{\boldsymbol{\theta}_t\}_{t=1}^{T}$ generated by DAT yields the convergence rate*

$$
\frac{1}{T} \sum_{t=1}^{T} \mathbb{E}\|\nabla_{\boldsymbol{\theta}}\Psi(\boldsymbol{\theta}_t)\|_2^2
$$
$$
= \mathcal{O}\left( \frac{1}{\sqrt{T}} + \frac{\sigma}{\sqrt{MB}} + \min\left\{ \frac{d}{4^b}, \frac{\sqrt{d}}{2^b} \right\} + \varepsilon \right), \quad (4)
$$

*where $b$ denotes the number of quantization bits, and $B = \min\{|\mathcal{B}_t^{(i)}|, \forall t, i\}$ stands for the smallest batch size per worker.*

**Proof**: Please see Appendix 3. $\square$

The error rate given by (4) involves four terms. The term $\mathcal{O}(1/\sqrt{MB})$ characterizes the benefit of using the large per-worker batch size $B$ and $M$ computing nodes in DAT. It is introduced since the variance of adaptive gradients (i.e., $\sigma^2$) is reduced by a factor $1/MB$, where $1/M$ corresponds to the linear speedup by $M$ machines. In (4), the term $\min\{\frac{d}{4^b}, \frac{\sqrt{d}}{2^b}\}$ arises due to the variance of compressed gradients, and the other two terms imply the dependence on the number of iterations $T$ as well as the $\varepsilon$-accuracy of the inner maximization oracle. We highlight that our convergence analysis (Theorem 1) is not barely a combination of LALR-enabled standard training analysis [You et al., 2019, 2017b] and adversarial training convergence analysis [Wang et al., 2019b, Gao et al., 2019]. Different from the previous work, we address the fundamental challenges in (a) quantifying the descent property of the objective value at the presence of multi-source errors during alternating min-max optimization, and (b) deriving the theoretical relationship between large data batch (across distributed machines) and the eventual convergence error of DAT.

# 6 EXPERIMENTS

We empirically evaluate DAT and show its success in training robust DNNs across multiple applications, which include ① adversarially robust ImageNet training, ② provably robust training by randomized smoothing, ③ semi-supervised robust training with unlabeled data, ④ robust transfer learning, ⑤ DAT using different communication protocols.

## 6.1 EXPERIMENT SETUP

**DNN models and datasets.** We use Pre-act ResNet-18 [He et al., 2016b] and ResNet-50 [He et al., 2016a] for image classification, where the former is shortened as ResNet-18. And we use ImageNet [Deng et al., 2009] for supervised DAT and augmented CIFAR-10 [Carmon et al., 2019] for semi-supervised DAT. In the latter setup, CIFAR-10 is augmented with unlabeled data drawn from 80 Million Tiny Images. When studying pre-trained model's transferability, CIFAR-100 is used as a target dataset for down-stream classification.

**Computing resources.** We train a DNN using $p$ computing nodes, each of which contains $q$ GPUs (Nvidia V100 or P100). Nodes are connected with 1Gbps ethernet. *A configuration of computing resources is noted by $p \times q$. If $p > 1$, then the training is conducted in a distributed manner.* And we split training data into $p$ subsets, each of which is stored at a local node. We note that the batch size $6 \times 512 = 3072$ is used for ImageNet over 36 GPUs. Unless specified otherwise, DAT is conducted using Ring-AllReduce, which requires one-sided quantization in Step 4 of Algorithm 1.

**Baseline methods.** We consider 2 *variants* of DAT: 1) *DAT-PGD*, namely, DAT using (iterative) PGD as the inner maximization oracle; and 2) *DAT-FGSM*, namely, DAT using one-step (projected) FGSM [Wong et al., 2020] as the inner maximization oracle. Additionally, we consider 4 training *baselines*: 1) *AT* [Madry et al., 2017]; 2) *Fast AT* [Wong et al., 2020]; 3) *DAT w/o LALR*, namely, a distributed implementation of AT or Fast AT but *without* considering LALR; and 4) *DAT-LSGD* [Xie et al., 2019], namely, a distributed implementation of large-batch SGD (LSGD) for AT. We remark that conventional AT and Fast AT are centralized training methods.

Table 1: DAT (in gray color) on (ImageNet, ResNet-50), compared with baselines, in TA (%), RA (%), AA (%), communication time per epoch (seconds), and total training time (including communication time) per epoch (seconds). For brevity, '$p \times q$' represents '# nodes $\times$ # GPUs per node', 'C' represents communication time in seconds, and 'T' represents training time in seconds.

| Method | ImageNet, ResNet-50 | | | | | | |
|---|---|---|---|---|---|---|---|
| | $p \times q$ | Batch size | TA (%) | RA (%) | AA (%) | C (s) | T (s) |
| AT | $1 \times 6$ | 512 | 62.70 | 40.38 | 37.46 | NA | 6022 |
| DAT-PGD w/o LALR | $6 \times 6$ | $6 \times 512$ | 57.09 | 34.02 | 30.98 | 865 | 1932 |
| DAT-PGD | $6 \times 6$ | $6 \times 512$ | 63.75 | 38.45 | 36.04 | 898 | 1960 |
| Fast AT | $1 \times 6$ | 512 | 58.99 | 40.78 | 37.18 | NA | 1544 |
| DAT-FGSM w/o LALR | $6 \times 6$ | $6 \times 512$ | 55.04 | 35.03 | 32.16 | 863 | 1080 |
| DAT-FGSM | $6 \times 6$ | $6 \times 512$ | 58.02 | 40.27 | 36.02 | 859 | 1109 |

**Training setting.** Unless specified otherwise, we choose the training perturbation size $\epsilon = 8/255$ and $2/255$ for CIFAR and ImageNet respectively, where recall that $\epsilon$ was defined in (1). We also choose 10 steps and 4 steps for PGD attack generation in DAT (and its variants) under CIFAR and ImageNet, respectively. The number of training epochs is given by 100 for CIFAR-10 and 30 for ImageNet. Such training settings are consistent with previous state-of-the-art [Zhang et al., 2019a, Wong et al., 2020]. To implement DAT-FGSM, we find that the use of cyclic learning rate suggested by Fast AT [Wong et al., 2020] becomes over-sensitive to the increase of batch size; see Appendix 4.2. Thus, we adopt the standard piecewise decay step size and an early-stop strategy [Rice et al., 2020] in DAT.

**Evaluation setting.** Unless specified otherwise, we report robust test accuracy (**RA**) of a learned model against PGD attacks [Madry et al., 2017]. Unless specified otherwise, we choose the perturbation size same as the training $\epsilon$ in evaluation, and the number of PGD steps is selected as 20 and 10 for CIFAR and ImageNet, respectively. We will also measure RA against AutoAttacks and the resulting robust accuracy is named **AA**. Further, we measure the standard test accuracy (**TA**) of a model against normal examples. All experiments are run 3 times with different random seeds, and the mean metrics are reported. We consider three different communication protocols, Ring-AllReduce (with one-sided quantization), parameter-server (with double quantization), and high performance computing (HPC) setting (without quantization). To measure the communication time, we use TORCH.DISTRIBUTED package with gloo and nccl as communication backend[1]. We then measure the time of required worker-server communications per epoch.

## 6.2 EXPERIMENT RESULTS

**Adversarial training on ImageNet.** In Table 1, we show an overall performance comparison on ImageNet between our proposed DAT variants and baselines in TA, RA, communication and computation efficiency. Note that AT and Fast AT are centralized training baselines using the same number of epochs as distributed training. **First**, we observe that the direct extension from AT (or Fast AT) to its distributed counterpart (namely, DAT-PGD w/o LALR or DAT-FGSM w/o LALR) leads to a large degradation of both RA and TA. **Second**, DAT-PGD (or DAT-FGSM) is able to achieve competitive performance to AT (or Fast AT) and enables a graceful training speedup. In practice, DAT is not able to achieve linear speed-up mainly because of the communication cost. For example, when comparing the computation time of DAT-PGD (batch size $6 \times 512$) with that of AT (batch size 512), the computation speedup (by excluding the communication cost) is given by $(6022)/(1960 - 898) = 5.67$, consistent with the ideal computation gain using $6\times$ larger batch size in DAT-PGD. **Third**, when comparing DAT-FGSM with DAT-PGD, we observe that the former leads to a larger loss in standard

---

[1] https://pytorch.org/docs/stable/distributed.html

accuracy (around 5%). Moreover, similar to Fast AT, we noted a larger RA variance of DAT-FGSM (around 1.5%) than DAT-PGD (around 0.5%). Thus, the FGSM-based training is less stable than the AT-based one, consistent with [Li et al., 2020].

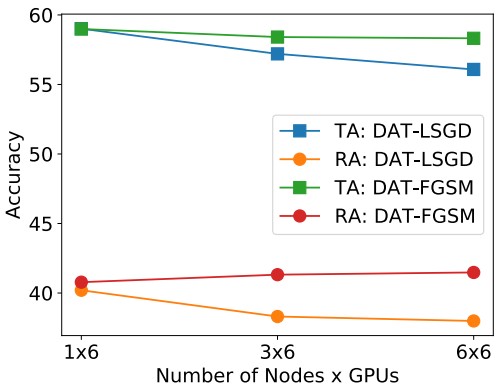

Figure 2: TA/RA comparison between DAT-FGSM and DAT-LSGD vs. node-GPU configurations on (ImageNet, ResNet-50).

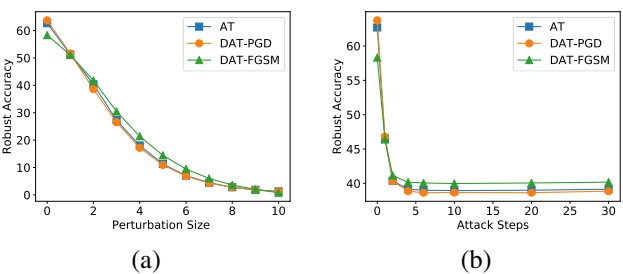

Figure 3: RA against PGD attacks for model trained by DAT-PGD, DAT-FGSM, and AT following (ImageNet, ResNet-50) in Table 1. (a) RA versus different perturbation sizes (over the divisor 255). (b) RA versus different steps.

In Figure 2, we further compare our proposed DAT with the DAT-LSGD baseline [Xie et al., 2019] in terms of TA/RA versus the number of computing nodes. Clearly, our approach scales more gracefully than the baseline, without losing much performance as the batch sizes increases along with the number of computing nodes. It is worth noting that our DAT setup is more challenging than [Xie et al., 2019], which used 128 GPUs but the per-GPU utilization is the 32 batch size. By contrast, although we only use 36 GPUs, the per-GPU batch size is 85. The use of a larger batch size per GPU makes distributed robust training useful when having access to a limited computing budget. Besides, Xie et al. [2019] used the PGD attack generation with a quite large number of attack steps (30) at the training time. This makes the computation time dominated over the node-wise communication time. However, we used a less number of PGD attack steps. In this scenario, the communication time cannot be neglected and prevents the practical

distributed implementation from achieving the linear speed-up. For example, when comparing the computation time of DAT-PGD with that of AT in Table 1, the computation speed-up (by excluding communication cost) is given by $6022/(1960 - 898) = 5.67$, close to the linear rate $(6\times)$.

In Figure 3, we evaluate the performance of DAT against PGD attacks of different steps and perturbation sizes (i.e., values of $\epsilon$). We observe that DAT matches robust accuracies of standard AT even against PGD attacks at different values of $\epsilon$ and steps.

**Adversarially trained smooth classifier.** DAT also provides us an effective way to speed up the smooth adversarial training (Smooth-AT) [Salman et al., 2020] for certified robustness. Different from AT, Smooth-AT augments a single training sample with *multiple* Gaussian noisy copies so as to train a Gaussian smoothing-aware classifier. Thus, Smooth-AT requires $N\times$ data batch size and storage capacity in contrast to AT, where $N$ is the number of noisy copies per sample. In the centralized training regime, $N$ can only be set by a small value, e.g., $N = 1$ or 2. However, DAT is able to scale up Smooth-AT with a large value of $N$, e.g., $N = 20$ in Table 2.

Table 2: Certified accuracy (%) of smooth classifiers trained by Smooth-DAT on (CIFAR-10, ResNet-18) versus $\ell_2$ radii. Here smooth classifiers are achieved at two Gaussian noise variance levels, $\sigma = 0.12$ and $\sigma = 0.25$, following [Salman et al., 2020]. And Smooth-AT is implemented using the baseline approach with $N = 2$ and the DAT approach with $N = 20$, respectively.

| Method | Smooth classifier ($\sigma = 0.12$) | | | | | | |
|---|---|---|---|---|---|---|---|
| | $r = 0.05$ | $r = 0.1$ | $r = 0.15$ | $r = 0.2$ | $r = 0.3$ | $r = 0.4$ | $r = 0.5$ |
| Baseline ($N = 2$) | 0.832 | 0.804 | 0.762 | 0.728 | 0.654 | 0.545 | 0 |
| DAT ($N = 20$) | 0.838 | 0.812 | 0.784 | 0.748 | 0.661 | 0.550 | 0 |
| Method | Smooth classifier ($\sigma = 0.25$) | | | | | | |
| | $r = 0.05$ | $r = 0.1$ | $r = 0.15$ | $r = 0.2$ | $r = 0.3$ | $r = 0.4$ | $r = 0.5$ |
| Baseline ($N = 2$) | 0.752 | 0.730 | 0.708 | 0.678 | 0.625 | 0.562 | 0.498 |
| DAT ($N = 20$) | 0.764 | 0.748 | 0.716 | 0.688 | 0.632 | 0.566 | 0.514 |

Smooth-AT can produce a provably robust classifier [Cohen et al., 2019]. To be more specific, let $f(\mathbf{x})$ denote a classifier (with input $\mathbf{x}$) trained by Smooth-AT. Then its Gaussian smoothing version, given by $f_{\text{smooth}}(\mathbf{x}) := \arg\max_c \mathbb{P}_{\boldsymbol{\delta}\in\mathcal{N}(\mathbf{0},\sigma^2\mathbf{I})}[f(\mathbf{x} + \boldsymbol{\delta}) = c]$, can achieve certified robustness, where $c$ is a class label, $\boldsymbol{\delta} \in \mathcal{N}(\mathbf{0}, \sigma^2\mathbf{I})$ denotes the standard Gaussian noise with variance $\sigma^2$, and $\mathbb{P}$ signifies the majority vote-based prediction probability over multiple noisy samples. The resulting smooth classifier $f_{\text{smooth}}$ can then be evaluated at certified accuracy [Cohen et al., 2019], a provable robust guarantee at a given $\ell_2$ perturbation radius $r$.

In Table 2, we present the certified accuracy (CA) of a $\sigma$-specified smooth classifier, obtained by either the conventional Smooth-AT approach (baseline using $N = 2$) or the DAT-enabled Smooth-AT method (DAT using $N = 20$).

And we evaluated CA at different $\ell_2$-radii. As we can see, DAT yields improved certified robustness over the conventional Smooth-AT with $N = 2$. This demonstrates the advantage of DAT in training provably robust classifiers: The use of a large number of Gaussian noisy samples becomes feasible through distributed training. We also observe that CA drops if the perturbation $\ell_2$-radius $r$ increases. This is not surprising since CA is derived by sanity checking if the certified $\ell_2$ perturbation radius of a smooth classifier can cover a given $r$. Besides, the smoother classifier constructed using Gaussian noises of large variance $\sigma$ tend to be more robust against a larger $\ell_2$ perturbation radius, but may hamper the accuracy against perturbations of small $\ell_2$ radius. This is consistent with [Cohen et al., 2019] and reveals the tradeoff between accuracy and certified robustness for a $\sigma$-specific smooth classifier.

Table 3: DAT with semi-supervision using ResNet-18 or Wide ResNet-28-10 under CIFAR-10 + 500K unlabeled Tiny Images.

| Method | ResNet-18, batch size $12 \times 2048$ | | | | |
| --- | --- | --- | --- | --- | --- |
| | TA (%) | RA (%) | AA (%) | C(s) | T(s) |
| DAT-PGD | 87.00 | 47.34 | 45.23 | 86 | 451 |
| DAT-FGSM | 88.00 | 45.84 | 43.19 | 86 | 124 |
| Method | Wide ResNet-28-10, batch size $12 \times 128$ | | | | |
| | TA (%) | RA (%) | AA (%) | C(s) | T(s) |
| DAT-PGD | 89.37 | 62.06 | 58.35 | 302 | 1020 |
| DAT-FGSM | 89.52 | 61.24 | 57.65 | 302 | 674 |

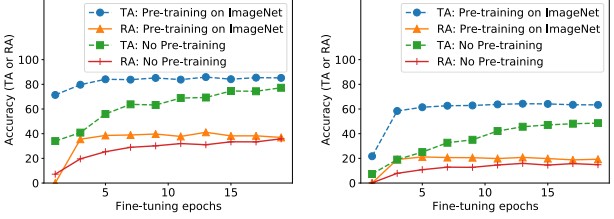

(a) Finetuning over CIFAR-10   (b) Finetuning over CIFAR-100

Figure 4: Fine-tuning ResNet-50 (pre-trained on ImageNet) under CIFAR-10 (a) and CIFAR-100 (b). For compassion, adversarial training on CIFAR datasets from scratch (no pretrain) is also presented. Here DAT-PGD is used for both pre-training and fine-tuning at 6 computing nodes.

**DAT under unlabeled data** In Table 3, we report TA and RA of DAT in the semi-supervised setting [Carmon et al., 2019] with the use of 500K unlabeled images mined from Tiny Images [Carmon et al., 2019]. As we can see, both DAT-PGD and DAT-FGSM scale well even if a $12\times$ batch size is used across 12 machines, each of which has a single GPU. We also compare DAT-PGD with [Carmon et al., 2019] following the latter's architecture, Wide ResNet 28-10. We note that DAT-PGD yields $89.37\%$ TA and $58.35\%$ AA. This is close to the reported $89.69\%$ TA and $59.53\%$ AA

in RobustBench [Croce et al., 2020] built upon small-batch adversarial training. Although the use of large data batch may cause a performance loss due to the reduced number of training iterations, the use of data augmentation serves as a remedy for such loss.

**DAT from pre-training to fine-tuning.** In Figure 4, we investigate if a DAT pre-trained model (ResNet-50) over a source dataset (ImageNet) can offer a fast fine-tuning to a down-stream target dataset (CIFAR-10/100). Here we up-sample a CIFAR image to the same dimension of an ImageNet image before feeding it into the pre-trained model [Shafahi et al., 2020]. Compared with the direct application of DAT to the target dataset (without pre-training), the pre-training enables a fast adaption to the down-stream CIFAR task in both TA and RA within just 3 epochs. Thus, the scalability of DAT to large datasets and multiple nodes offers a great potential to rapidly initialize an adversarially robust base model in the 'pre-training + fine-tuning' paradigm.

**Quantization effect in various communication protocols** In Table 4, we present how DAT is affected by gradient quantization. As we can see, when the number of bits is reduced, the communication cost and the amount of transmitted data are saved, respectively. However, the use of an aggressive gradient quantization introduces a performance loss. For example, compared with the case of using 32 bits, the most aggressive quantization scheme (8-bit 2-sided quantization in Steps 4 and 7 of Algorithm 1) yields an RA drop around $4\%$ and $7\%$ for DAT-PGD and DAT-FGSM, respectively. In particular, DAT-FGSM is more sensitive to the effect of gradient quantization than DAT-PGD. It is worth noting that our main communication configuration used in previous experiments is Ring-AllReduce that calls for 1-sided (rather than 2-sided) quantization. We further show that if a high performance computing (HPC) cluster of nodes (with NVLink high-speed GPU interconnect [Foley and Danskin, 2017]) is used, the communication cost can be further reduced without causing performance loss.

# 7 CONCLUSIONS

We proposed distributed adversarial training (DAT) to scale up the training of adversarially robust DNNs over multiple machines. We showed that DAT is general in that it enables large-batch min-max optimization and supports gradient compression and different learning regimes. We proved that under mild conditions, DAT is guaranteed to converge to a first-order stationary point with a sub-linear rate. Empirically, we provided comprehensive experiment results to demonstrate the effectiveness and the usefulness of DAT in training robust DNNs with large datasets and multiple machines. In the future, it will be worthwhile to examine the speedup achieved by DAT in the extreme training cases, e.g., using a significantly large number of attack steps and

Table 4: Effect of gradient quantization on the performance of DAT for various numbers of bits. The training and evaluation settings on (ImageNet, ResNet-50) are consistent with Table 1. The new performance metric 'Data trans. (MB)' represents data transmitted per iteration in the unit MB.

| Method | ImageNet, ResNet-50 | | | | |
|---|---|---|---|---|---|
| | # bits | TA (%) | RA (%) | C (s) | Data trans. (MB) |
| DAT-PGD | 32 | 63.75 | 38.45 | 898 | 2924 |
| DAT-PGD | 16 | 61.77 | 38.40 | 850 | 1462 |
| DAT-PGD | 8 | 56.53 | 37.90 | 592 | 731 |
| DAT-PGD | 8 (2-sided) | 53.09 | 34.59 | 1091 | 244 |
| DAT-PGD (HPC) | 32 | 63.43 | 38.55 | 15 | 1074 |
| DAT-FGSM | 32 | 58.02 | 40.27 | 859 | 2924 |
| DAT-FGSM | 16 | 54.71 | 39.29 | 849 | 1462 |
| DAT-FGSM | 8 | 50.11 | 36.38 | 594 | 731 |
| DAT-FGSM | 8 (2-sided) | 48.27 | 33.20 | 1013 | 244 |
| DAT-FGSM (HPC) | 32 | 57.60 | 41.70 | 15 | 310 |

distributed machines, and extremely large models.

## Acknowledgements

Y. Zhang and S. Liu are supported by the Cisco Research grant CG# 70614511. P. Khanduri and M. Hong are supported in part by the NSF grants 1910385 and 1727757. We also thank Dr. Cho-Jui Hsieh for the helpful discussion on early ideas of this paper.

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
