# OpenReview forum: "Distributed Adversarial Training to Robustify Deep Neural Networks at Scale"
_auai.org/UAI/2022/Conference — UAI 2022 Oral_

### Official Review · Reviewer_fQMJ · 2022-03-23

**Q2(1) Originality/Novelty:** 3
**Q2(2) Significance/Impact:** 4
**Q2(3) Correctness/Technical Quality:** 3
**Q2(6) Clarity Of Writing:** 4
**Q6 Overall Score:** 8
**Q8 Confidence In Your Score:** 3

**Q1 Summary And Contributions:**

This paper proposes an adversarial training method under the distributed learning setting.
The main obstacle is the large-batch problem, known in the distributed learning community, but not handled with adversarial training yet.
The authors integrate the layerwise adaptive learning rate scheduling, for which the convergence result is successfully given.
The extensive experimental evaluation witnesses the significance of the proposed method.

**Q2 Assessment Of The Paper:**

More detailed information regarding each of these aspects is given below:

**Q2(4) Quality Of Experiments (Optional):**

4: Excellent: The experimental evaluation is comprehensive and the results are compelling.

**Q2(5) Reproducibility:**

3: Good: Key resources (e.g., proofs, code, data) are available and key details (e.g., proofs, experimental setup) are sufficiently well-described for competent researchers to confidently reproduce the main results.

**Q3 Main Strengths:**

The problem handled in this paper is an important one in the machine learning community---scaling up adversarial training.
As a result, the authors propose a simple learning rate scheduling, which is practically simple to implement and admits the convergence result at the same time.

The experimental evaluation is thorough.
The authors do not only simulate that the proposed learning rate scheduling admits the better robustness and accuracy but also study the hyperparameter sensitivity in many ways, including the perturbation size, attack steps, inner loop oracles, etc.
Such experimental results ensure the practical usefulness in a convincing way.

**Q4 Main Weakness:**

I don't see any major weaknesses, except for minor ones mentioned in Q5.

**Q5 Detailed Comments To The Authors:**

- Can you describe what is AutoAttacks in "Evaluation setting" in Sec. 6.1?

- Can you explain how Smooth-AT (used in Sec. 6) is incorporated in the formulation of equation (2) in detail?

**Q7 Justification For Your Score:**

The adversarial training under the distributed learning has practical importance, and the proposed method is simple yet very effective as shown in theory and experiments.
These are sufficient reasons for the acceptance.

**Q9 Complying With Reviewing Instructions:**

1: Yes.

---

### Official Review · Reviewer_dV5C · 2022-04-13

**Q2(1) Originality/Novelty:** 4
**Q2(2) Significance/Impact:** 4
**Q2(3) Correctness/Technical Quality:** 3
**Q2(6) Clarity Of Writing:** 4
**Q6 Overall Score:** 8
**Q8 Confidence In Your Score:** 3

**Q1 Summary And Contributions:**

To scale up training, the authors adapt adversarial training to the distributed context. The contributions about Distributed Adversarial Training (DAT) are :
(i) DAT supports various types of attacks
(ii) gradient compression operations can be employed in this framework
(iii) under standard assumptions, DAT is theoretically shown to converge to the first-order stationary points in general non-convex settings.
(iv) DAT compares to or outperforms state-of-the-art  approaches and speeds up training

**Q10 Ethical Concerns (Optional):**

No Ethical Concerns

**Q2 Assessment Of The Paper:**

More detailed information regarding each of these aspects is given below:

**Q2(4) Quality Of Experiments (Optional):**

3: Good: The experimental evaluation is adequate, and the results convincingly support the main claims.

**Q2(5) Reproducibility:**

3: Good: Key resources (e.g., proofs, code, data) are available and key details (e.g., proofs, experimental setup) are sufficiently well-described for competent researchers to confidently reproduce the main results.

**Q3 Main Strengths:**

The contribution is original. There is currently a large body of work on how to train deep neural networks to be robust against adversarial attacks. None of the proposals is satisfactory so far :  the approach by Madry and collaborators (2017) and further work in this line are an exception ; however, they are computationally demanding and are difficult to scale.

The contributions are well described.

A theoretical quantification on the effect of descent errors from multiple sources  on the convergence of DAT is provided.

DAT resorts to large-batch optimization to improve training scalability.

A comprehensive experimental study successively addresses the capability of DAT to robust training
- over ImageNet,
- by randomized smoothing,
- with unlabeled data,
- across various computing and communication settings.

**Q4 Main Weakness:**

I can see none.

**Q5 Detailed Comments To The Authors:**

1) page 3 : Section 4 Methodolog<<<ies>>>. I do not see the necessity to use plural here.

**Q7 Justification For Your Score:**

See Q3

**Q9 Complying With Reviewing Instructions:**

1: Yes.

---

### Official Review · Reviewer_Pngd · 2022-04-17

**Q2(1) Originality/Novelty:** 1
**Q2(2) Significance/Impact:** 1
**Q2(3) Correctness/Technical Quality:** 3
**Q2(6) Clarity Of Writing:** 3
**Q6 Overall Score:** 5
**Q8 Confidence In Your Score:** 3

**Q1 Summary And Contributions:**

The authors propose a large-batch distributed adversarial training (DAT) framework for robustness. They analyze the convergence of DAT to first-order stationary points with sub-linear rate. Also they show experimental results to verify their proposed method with various practical settings.

**Q2 Assessment Of The Paper:**

More detailed information regarding each of these aspects is given below:

**Q2(4) Quality Of Experiments (Optional):**

3: Good: The experimental evaluation is adequate, and the results convincingly support the main claims.

**Q2(5) Reproducibility:**

3: Good: Key resources (e.g., proofs, code, data) are available and key details (e.g., proofs, experimental setup) are sufficiently well-described for competent researchers to confidently reproduce the main results.

**Q3 Main Strengths:**

- The proposed method and the analysis for the convergence rate of DAT are reasonable.
- They covered various variants of AT (e.g., semi-supervised) and showed its practicality.


**Q4 Main Weakness:**

- The contribution for this paper is not novel, most techniques are combined by using previous works (e.g., Large batch training, Gradient quantization)
- The analysis of convergence for DAT are reasonable, but it seems the similar approach had been proposed before [1]. What is the novelty for the proposed analysis?
- I think the empirical experiments are too limited. Is it still valid for other methods (e.g., AutoAttack) which requires more computations or an extremely large number of attack steps?
- All empirical experimental results would be better if there are more random seeds setting. (e.g., Table 1 shows that there is no significant difference between Fast AT and DAT)
[1] Wang, Y., Ma, X., Bailey, J., Yi, J., Zhou, B., & Gu, Q. (2021). On the convergence and robustness of adversarial training.

**Q5 Detailed Comments To The Authors:**

- In the experimental setting, the author did not adopt the same learning strategy (cyclic lr) for the Fast-AT. I’m concerned because the baseline seems to be slightly different.
- I think it is necessary to clearly explain the advantages and novelty of the proposed method compared to the existing other fast AT technique [2].
[2] DONG, Yinpeng, et al. Adversarial distributional training for robust deep learning. Advances in Neural Information Processing Systems, 2020, 33: 8270-8283.

**Q7 Justification For Your Score:**

The convergence theoretical analysis is not novel, similar approaches had been proposed before.


**Q9 Complying With Reviewing Instructions:**

1: Yes.

---

### Decision · Program_Chairs · 2022-05-15

**Decision:**

Accept (Oral)

**Comment:**

Meta Review: This paper was judged as very strong by two reviewers and the third was also reasonably enthusiastic. An important problem is addressed in a novel and effective manner.